# Stem Cell Therapy in Children with Traumatic Brain Injury

**DOI:** 10.3390/ijms241914706

**Published:** 2023-09-28

**Authors:** Wen-Ya Lin, Kang-Hsi Wu, Chun-Yu Chen, Bei-Cyuan Guo, Yu-Jun Chang, Tai-An Lee, Mao-Jen Lin, Han-Ping Wu

**Affiliations:** 1Department of Pediatrics, Taichung Veterans General Hospital, Taichung 40705, Taiwan; wylin002@gmail.com; 2Department of Pediatrics, Chung Shan Medical University Hospital, Taichung 40201, Taiwan; cshy1903@gmail.com; 3School of Medicine, Chung Shan Medical University, Taichung 40201, Taiwan; 4Department of Emergency Medicine, Tung’s Taichung MetroHarbor Hospital, Taichung 433, Taiwan; yoyo116984@gmail.com; 5Department of Nursing, Jen-Teh Junior College of Medicine, Nursing and Management, Miaoli 79-9, Taiwan; 6Department of Pediatrics, National Cheng Kung University Hospital, College of Medicine, National Cheng Kung University, Tainan 704, Taiwan; gbc628@gmail.com; 7Laboratory of Epidemiology and Biostastics, Changhua Christian Hospital, Changhua 500, Taiwan; 83686@cch.org.tw; 8Department of Emergency Medicine, Chang Bing Show Chwan Memorial Hospital, Changhua 505, Taiwan; d0i6a0n9e@hotmail.com; 9Division of Cardiology, Department of Medicine, Taichung Tzu Chi Hospital, The Buddhist Tzu Chi Medical Foundation, Taichung 427413, Taiwan; 10Department of Medicine, College of Medicine, Tzu Chi University, Hualien 970, Taiwan; 11College of Medicine, Chang Gung University, Taoyuan 33302, Taiwan; 12Department of Pediatrics, Chiayi Chang Gung Memorial Hospital, Chiayi 613, Taiwan

**Keywords:** stem cell therapy, traumatic brain injury, children, mesenchymal stem cells, bone marrow mononuclear cells

## Abstract

Pediatric traumatic brain injury is a cause of major mortality, and resultant neurological sequelae areassociated with long-term morbidity. Increasing studies have revealed stem cell therapy to be a potential new treatment. However, much work is still required to clarify the mechanism of action of effective stem cell therapy, type of stem cell therapy, optimal timing of therapy initiation, combination of cocurrent medical treatment and patient selection criteria. This paper will focus on stem cell therapy in children with traumatic brain injury.

## 1. Introduction

Pediatric traumatic brain injuries cause major mortality and morbidity. The current treatment regime has improved survival, but specific measures aiming to reduce neurological damage and subsequent chronic disability still require exploration. Much preclinical animal studies and few specific clinical studies have been performed to explore the pathogenesis of traumatic brain injury and the regenerative effect of stem cells in the treatment of traumatic brain injury. A newer generation of stem cell therapy including the role of secretomes and exosomes is also under investigation.

Searches in “PubMed” were conducted, with the initial manuscript search restricted to those written within the last 10 years. Emphasis was placed on selecting the most recent and comprehensive papers. The most recent manuscripts were employed with those topics without relevant information found within the last 10 years. This paper attempts to provide a brief narrative and clinical review of the role of stem cell therapy in pediatric traumatic injury.

## 2. Stem Cells

Stem cells (SCs) provide the basic structural unit of tissues and organs; they have a unique ability to self-regenerate and differentiate into multiple cell lineages [1]. Characteristics of SCs include clonality (they usually arise from a single cell) and potency (ability to differentiate into various cells and tissues). Stem cells can thus be classified according to origin of extraction and potency (Table 1) [2]. The proliferation and differentiating capability of SCs may differ according to the source of origin. Mesenchymal stem cells (MSC) derived from umbilical cord-derived Wharton’s jelly had the strongest proliferative and differentiation potential [3]. Depending on their potency, five classes of cells can be identified. These range from the most undifferentiated to the most differentiated cell types, including omnipotent, pluripotent, multipotent, oligopotent and unipotent stem cells. An example of the multipotent stem cell type is mesenchymal stem cells; they can differentiate into mesoderm-derived tissues such as adipose tissue, bone, cartilage and muscle; they also exhibit trans-differentiation, forming neuronal tissue which is of ectodermal origin [4]. Induced pluripotent stem cells (iPSCs) are a new advancing source of stem cells. The relevant technique allows oligopotent or unipotent somatic cell to be reprogrammed to a less differentiated pluripotent state [5]. Thereafter, directed differentiation via mimicking the microenvironment and extracellular signals allows these pluripotent cells to be converted to desired cell types [6].

Stem cell therapy elicits a complex regenerative response and may display variable activity in the presence of different stimuli and microenvironments. Different to the traditional oral route of drug administration, oral passage into the gastrointestinal system is not favored for effective stem cell therapy. Other routes, such as intravenous, intra-arterial, local and intrathecal, have been suggested [7] (Figure 1). The intravenous route for stem cell application for hematological diseases such as leukemia has been well established [8]. However, the advised routes of administration may differ, depending on the underlying specific disease. The pharmacokinetics of stem cell therapy also needs to be determined, including their biodistribution after infusion, survival of engrafted cells and incorporation into areas of damage [7]. Thus, in utilizing stem cell therapy, various factors, such as route, timing and dose of administration; donor factor; host factor; ex vivo differentiation; and appropriate autologous or allogenic therapy, are vital to optimize treatment efficacy [9].

## 3. Traumatic Brain Injury

Traumatic brain injury (TBI) in children may result in mortality or permanent disability. The 2013 US CDC data revealed there were more than 640,000 TBI-related emergency department visits for children aged 14 years or younger. Pediatric TBI visit to an emergency department increased between 2007 and 2013 for the 0–4-year and 5–14-year age group, rising by 37.8% in the youngest age group [10].

TBI can be classified as an open or closed injury, according to the nature of sustained trauma. Most of the injuries are of the closed type. Traumatic damage to the brain consists of two phases and can be divided into primary and secondary injury. Primary injury is due to direct mechanical force with immediate damage to intracranial content. Secondary injury arises from subsequent response (altered cerebral blood flow and inflammation) to initial damage and takes place over hours to days [11]. Vasospasm, focal microvascular occlusion and vascular injury alter cerebral blood flow, because the blood–brain barrier is compromised, and the loss of this barrier function allows for the passage of immune cells and foreign elements (therapeutic and neurotoxic) into the traumatized brain [12]. Cytotoxic- and vasogenic-related cerebral edema contribute to raised intracranial pressure and further compromised blood flow; this secondary ischemia leads to hypoxia and progressive neuronal cell death. This disruption of cerebral blood flow occurs within 24 h of trauma with resultant tissue ischemia [13].

With TBI, children may initially present with vomiting, blurred vision, headache, disorientation and gait abnormalities (Figure 2). The clinical severity of TBI is often classified by the Glasgow coma scale (GCS). The GCS evaluates individual responses according to visual, verbal and motor function. Response to stimuli is converted into an appropriate scoring system. These include best visual response with spontaneous eye opening (maximum score of 4), best orientated verbal response (maximum score of 5) and best appropriate motor response to command (maximum score of 6). The total sum of these scores yields a maximum score of 15 and minimum score of 3. Severe TBI corresponds to a GCS score of 3–8, moderate TBI to 9–12 and mild TBI to 13–15 [14]. Moderate to severe TBI may be fatal or cause long-term disability.

Clinical presentation is variable and dependent on the severity and nature of trauma and the age of patient [15]. Subsequent disability with physical, cognitive, emotional and behavioral deficits may have prolonged influence for many years after the initial insult. Various models have been proposed to explain recovery from early-injured brain. Models of early vulnerability propose injuries to still immature brain: thus, cognitive, social and behavioral deficits may only be apparent in later adolescence or adulthood, when the cortical brain region and associated skills are expected to reach maturity [16]. In contrast, the early plasticity perspective proposes that the immature brain is less structurally refined, and is able to reorganize effectively with limited functional loss in response to traumatic injury [17,18,19]. Other influences, including a child’s preinjury characteristics, family environmental factors and brain developmental stage, may also affect the final chronic neurobehavioral outcome [16]. A child’s preinjury cognitive reserve is an important determining factor; those with a higher preinjury intelligence quotient (IQ) score and adaptive function have a more favorable cognitive outcome after injury [20]. Family environmental factors such as belonging to a low socioeconomic status, having a lack of resources and experiencing family stress may have negative effects. Parental negativity may hinder behavioral recovery after TBI [21]. Although these models may serve as useful frameworks in predicting chronic outcome, the actual clinical presentation is far more complex and variable.

Chronic neurological outcomes in pediatric TBI include impairment in intellectual functioning, information processing, memory, learning, executive functioning and social and behavioral outcomes. Executive functioning refers to higher-order skills involved in execution; these may include working memory, inhibitory control, self-regulation, organization and decision making [16]. Long-term adverse influences on social and behavioral outcome have also been reported. School-aged children with TBI have a higher risk of impaired socialization, communication and adaptive behavior [22,23]. These children have increased depressive symptoms, anxiety, aggression and antisocial behavior [24,25].

Recovery is determined by severity of injury, time since injury and developmental stage at time of injury. A total of 181 children were categorized into four groups according to age when TBI was sustained; these were infant, preschool, middle childhood and late childhood. It was found that children with TBI sustained in middle childhood had lower IQ scores evaluated 2 years after the initial injury [26]. Severe TBI in children is associated with poorer recovery of intellectual ability [27]. Poorer academic function and the worst executive function score were found in children with severe TBI; those with mild to moderate TBI also had some memory deficits [28,29].

Current two-staged treatments include initial repair of primary injury and immediate consequence; these may include drainage of intracranial bleeding, debridement of nonviable tissue and achieving hemostasis. Subsequent interventions were aimed to decrease intracranial pressure and minimize ongoing secondary neuroinflammation [11]. Guidelines specifically for the treatment of severe pediatric traumatic brain injury were recommended. The first component targets three therapeutic goals, namely, preventing and/or treating raised intracranial pressure, measures to optimize cerebral perfusion pressure and partial pressure of braintissue oxygen. The second component of therapy includes barbiturate infusion, late application of moderate hypothermia, induced hyperventilation and hyperosmolar therapies [30].

## 4. Stem Cell Therapy and Traumatic Brain Injury

Treatment for TBI has improved survival rate; however, clinical treatment for functional brain recovery and minimizing neurological sequelae is still lacking. Stem cells have the ability to prevent cell death and play a neuroprotective role in tissue reconstruction. Mesenchymal and neural stem cells may serve as potential new treatments [31].

### 4.1. Neural Stem Cell Response to Injury

Neural stem cells (NSCs) and neural progenitor cells (NPs) reside in the germinal zone of immature brain. Interest in their regenerative potential has been raised; the brain subventricular zone (SVZ) comprises a mixture of primitive neuroepithelial cells in the embryonic ventricular zone and intermediate progenitors assortments [32]. Neural stem cells are multipotential, and NPs are either multipotential, bipotential or unipotential [33]. Generated immature neurons migrate in tangentially oriented chains, with eventual convalescence into cell streams at the selected designation. Studies were carried out in human and rodent models. These demonstrate six main streams of progenitor cells arising from the SVZ during postnatal development (Table 2) [34,35,36,37,38,39]. The SVZ has the ability to generate neurons; however, functions are mainly gliogenic, generating initially astrocytes and later oligodendrocytes [40].

In contrast to animal studies, the structure and activity of the SVZ in human infants differ from those of adulthood; this change is more apparent after 6 months of age. Between 6 and 18 months of age, a gradual depletion of the migratory neuron network is found, with progression to the adult structure with an astrocyte ribbon and hypocellular gap layer. Doublecortin (DCX) is a protein expressed by immature migrating neuroblasts. With increasing age, a decline inimmature neurons expressing DCX neural markers was found. This is further echoed by a study of neurogenesis which showed no associated increased DCX+ cells in pediatric TBI patients. In addition, in comparison to younger children aged 2 to 6 years and older children aged 7 to 10 years, the density of immature migrating neuroblasts was significantly greater in infants younger than 1 year of age [37]. In the adult human brain, the productions of olfactory bulb neurons from SVZ progenitor cells are not sustained throughout life. This production is no longer detectable after 2 years of age [41]. This suggests the subsidence of SVZ neurogenesis, with near abolishment by adulthood and restricted response to TBI [35]. Thus, recovery from TBI is limited by the irreversible loss of neurons, limiting self-regeneration with resulting neurological deficits. This is demonstrated in a clinical study of 45 children with TBI sustained at ages ranging from 3 to15 years of age. Severe functional impairment was noted within the first 3 months. Despite a more significant clinical recovery over the first year, most children still had persistent impairments at 1 year after TBI, and the majority of children (80%) had persistent moderate to severe disability at the 2-year follow-up [42]. Dentate gyrus is important in learning and memory.

### 4.2. Neuroinflammation Response to Injury

Following pediatric TBI, prolonged neuroinflammation and dysregulation result in neuronal loss in the hippocampal dentate gyrus and associated cognitive deficits [43]. Neuroinflammation following injury also influences TBI outcome. This is due to the interplay between neurons, glial cells and soluble cytokines and chemokines [44].

Activated microglia are resident macrophages in the brain; they play in role in regulating this neuroinflammation process. Their interactions with neurons determine the specific type of chemokines and cytokines secreted and thus the eventual function exerted [45]. A mice model study has yielded a neuroprotective effect after inducing microglial turnover in TBI-injured brain. This beneficial effect is dependent on interleukin-6 trans-signaling and enhances survival of newborn neurons that favor cognitive function [46]. After TBI, astrocytes secrete both proinflammatory cytokines (interleuking-1β) and neurotrophic factors (transforming growth factor beta). These reactive astrocytes are important in pediatric TBI, with excessive activation resulting in increased inflammatory response, leading to glial scar formation and limiting neural regeneration and neuroplasticity [47]. Oligodendocyte progenitor cells (OPCs) mature into oligodendrocytes (OLGs) and play a role in myelination. TBI with traumatic axonal injury can lead to demyelination in axons and death of OLGs. Following TBI, proinflammatory cytokines can trigger death of OPCs and OLGs, resulting in progressive white matter injury and chronic functional deficits [48,49].

Multiple cytokines are produced in response to brain injury; these include erythropoietin, interleukin-6, notch receptors such as delta/Jagged/Notch-1, leukemia inhibitory factors, transforming growth factor-α, vascular endothelial cell growth factor-A and-C and transforming growth factor-β1 (TGF-β1) [32]. Some exert a positive effect and favor the production of neurons. The notch receptors increase the number of proliferating NSPs in the SVZ [50]. The administration of erythropoietin enhances neurogenesis and oligodendrogenesis and reduces the number of astrocytes produced [51]. Other cytokines such as TGF-β1 may inhibit neural progenitor cell proliferation and thus prevent neurogenesis in the brain [32,52].

Persistent inflammation, cell damage and neuronal dysfunction may be found after initial primary damage and may extend for years. The initial inflammation in acute brain injury may be protective; however, aberrant persistent inflammation thereafter may lead to secondary cell death and chronic disability. This shift from protective to degenerative inflammation is complex and serves as potential target for the combination of stem cell and anti-inflammatory therapy [53].

Thus, in human TBI, the capacity of neural self-repair is more restricted, with cell death favored, especially in the presence of a highly inflammatory environment following injury [31]. Mesenchymal stem cells possess the ability to migrate to the site of injury, secrete anti-inflammatory protein and modulate cellular effectors of inflammatory response [53]. Thus, MSCs may aid in this self-limited regenerative process.

### 4.3. Stem Cell Therapy

A wide variety of neuronal cell types may be required to restore neurological function in a selected patient with moderate to severe TBI. Neural stem cells (embryonic or adult) self-renew and differentiate into neurons and glial cells, replacing damaged neurons and promoting neurogenesis in injured brain [54]. This is conceptually attractive, but a few issues need to be overcome before actual clinical application. These NSCs do not migrate far beyond the area of delivery; thus, multiple stereotactic injections may be required, and this still does not guarantee SC delivery to the required injury site [55]. Thesource of NSCs’ harvest is an ethical issue, and there is also the risk of transplant rejection. These factors limit the clinical applications of NSCs.

Other stem cell types, including multipotent adult progenitor cells and endothelial progenitor cells, have also been studied in TBI. However, their safety and efficacy is not yet conclusive [56]. Induced pluripotent cells (iPSCs) are more appealing, as they can be harvested from patients themselves, reprogrammed in vitro and then delivered back to the patients. This avoids possible ethical problems and immune rejection [56]. Induced pluripotent cells have been shown to improve cognition and motor performance in animal studies [57]. In its reprogramming process, infection with virus is used; concern over possible tumorigenicity has been expressed. Thus, the safety of iPSCs requires more determination before clinical use.

The MSCs have been applied in some clinical studies and show more promise for clinical application. These cells are multipotent and can be harvested from bone marrow, umbilical cord, adipose tissue, placenta and a variety of tissue types. Mesenchymal stem cells cross the blood–brain barrier and travel to the site of injury, release trophic factors including brain-derived neurotrophic factor and nerve growth factor, recruit local progenitor cells and modulate appropriate anti-inflammatory response to reduce cell death and favor repair [53,54,58]. Mesenchymal stem cells derived from different sources also have the ability to differentiate into various cell types. Those derived from bone marrow can differentiate into several cell lineages, including neurons and glial cells [59]. Adipose tissue-derived MSCs can differentiate into neurons, endothelial-derived cells and Schwann cells [60]. Umbilical cord- or cord blood-derived MSCs allow easier extraction and have low immunogenicity power and less risk of rejection.

Table 3 reveals published clinical human studies specific to TBI; these are few, but showed promising results [61,62,63,64]. Liao et al. performed a study in children between age 5 and 14 years and with a GCS score of 5–8; intravenous autologous bone marrow mononuclear cell therapy (dose of 6 × 10^6^ stem cells per kilogram of body weight) was administered within 48 h of injury. Ten children received treatment and nineteen children were assigned to the control group. The primary outcome was evaluated by using the Pediatric Intensity Level of Therapy scale to quantify treatment intensity of elevated intracranial pressure. Secondary outcomes were measured by utilizing the Pediatric Logistic Organ Dysfunction score and days of intracranial pressure monitoring. Children treated with cocurrent stem cell therapy had a less intensive treatment regime for intracranial pressure management, with significant reduction beginning at 24 h post-treatment to within the first week of treatment. Reduced organ injury severity and decreased duration of neurointensive care were also reported. These findings demonstrated that MSCs were able to reduce the inflammation effect in acute TBI and initial acute treatment intensity and duration [61]. This study also revealed the harvesting of autologous MSCs in acute injury to be safe and feasible [55].

In an extension from the same studied group, Cox et al. further demonstrated favorable neuropsychological and functional outcomes in 10 children receiving autologous bone marrow mononuclear cell therapy at 6 months’ follow-up. Tools used to evaluate functional outcome (Glasgow outcome scale, Glasgow outcome scale—expanded for children, pediatric injury functional outcome scale and adaptive behavior assessment system II) all showed significant improvement. A significantly improved neuropsychological outcome was measured with the Wechsler abbreviated scale of intelligence, coding, grooved pegboard, listening recall and verbal learning. Magnetic resonance imaging also revealed no progressive post-TBI brain tissue loss [62].

Similarly, in a study of 40 adults (20 treatment patients and 20 control patients) with sequelae of TBI sustained more than 1 year ago, the transplantation of umbilical mesenchymal stem cells (dose of 1 × 10^7^ stem cells × 4 courses in 5 to 7 days) delivered with lumbar puncture in the lumbar 3–4 and 4–5 intervertebral spaces had revealed improved neurological function and self-care. A Fugl-Meyer assessment and functional independence measure scores were assessed at baseline and 6 months after treatment; patients with stem cell therapy showed significant improvement in motor function, sensitivity and balance [63].

Tian et al. conducted an interventional cohort study in 97 patients with serious TBI for at least 1 month. Among the 97 enrolled patients, 24 had persistent vegetative state and 73 had disturbances in motor activity. Intrathecal administration of autologous bone marrow stem cells (dose of 1 × 10^6^ stem cells) was applied. Significant improvement in brain function was found in 38 of 97 patients (*p* = 0.007), improved consciousness was found in 11 of 24 patients with persistent vegetative state (*p* = 0.024) and improved motor function was found in 27 of 73 patients with disturbances in motor activity (*p* = 0.025). More prominent improvement was noted with younger patients, possibly due to the bodies’ better baseline condition in the younger age group. An inverse relationship between time after injury and treatment outcome was found. Thus, an earlier initiation of stem cell therapy in the subacute stage of traumatic brain injury confers a better result [64].

All these studies reported no major adverse effects. Minor discomfort such as transient fever, dizziness and headache after lumbar puncture all resolved without long-term effects. These demonstrated safe application of mesenchymal stem cell therapy in both adult and pediatric patients with TBI.

Current ongoing clinical trials of stem cell therapy in TBI are listed by Schepici et al. [65]. However, to the authors’ knowledge, no published results were available at the time of writing of this manuscript.

## 5. Challenges and Future Prospects

Mesenchymal stem cells possess features that make them a potentially promising clinical treatment option for children with TBI (Table 4). MSCs are multipotent and can differentiate into a variety of cell types, including those of neural lineage [66,67]. They are able to migrate to target the site of injury, which is mainly facilitated by various chemotactic factors, the most potent influence coming from platelet-derived growth factor-AB (PDGF-AB) and insulin-like growth factor 1 (IGF-1) [68]. Mesenchymal stem cells express molecules responsible for leucocyte tethering, rolling and transmigration processes and are thus able to migrate from blood and cross endothelial cells into damaged tissues. These include various integrins, selectins and vascular adhesion molecules [69]. Specific to TBI, MSCs are able to cross the blood–brain barrier (BBB), using the above-mentioned mechanism [70]. In addition, other means of migration included passage through transiently formed interendothelial gaps between brain microvascular endothelial cells [71]. Invasion to local tissues via plasmic podia has also been described [72]. Mesenchymal stem cells and secreted cytokines promote regeneration in damaged tissue, stimulating the proliferation, migration and differentiation of surviving endogenous neural stem cells at the site of damage [73]. Mesenchymal stem cells also have immunosuppressive properties; this may also help to minimize adverse effects of TBI-related secondary injury [12].

Past explorations of stem cell therapy in TBI were predominantly focused on animal studies, with the few studies on human TBI revealing promising results. An increasing number of recent studies have also revealed feasible clinical application of MSC therapy in various other pediatric diseases [9]. This includes the application of bone marrow-derived MSCs in treating bone disorders such as osteogenesis imperfecta, graft-versus-host disease, lysosomal storage disease and spinal muscular atrophy. Umbilical cord MSCs were also applied in patients with bronchopulmonary dysplasia, cerebral palsy and autism spectrum disorders.

Cerebral palsy (CP) results from non-progressive damage to developing brain, and TBI can be one of the causes. Stem cell therapy applied to children with CP seems to be effective and safe. Case reports on cohort studies have reported treatment with umbilical cord blood stem cell transplantation (months to years after initial insult) with improved neurological outcome [74]. These findings are further echoed by a meta-analysis performed by Qu et al.; MSCs derived from bone marrow and other sources were applied, all with some improved neurological recovery and minimal adverse effect at 2 years’ follow-up. These results are promising. However, cautious interpretation should be applied. There were only nine studies included in this meta-analysis with high heterogenicity. Thus, further high-quality randomized control trials should be performed [75].

Mesenchymal stem cell therapy is a safe and feasible treatment option for children with TBI. A brief flowchart of MSC therapy is proposed in Figure 3. Treatment in acute injury can be offered as combination therapy with the current medical or surgical regime, aiming at decreasing intracranial pressure, modulating acute inflammatory response and decreasing duration and intensity of initial critical neuroinvasive care. Treatment beyond the acute phase can possibly regulate chronic secondary injury, promote neurological recovery and minimize neurological sequelae. This is especially important in children, as TBIs lead to damage to immature brain; thus, timely regenerative stem cell therapy should be offered to minimize chronic neurological disability.

In designing an MSC therapy model, various issues need to be considered (Table 4). There are multiple sources of mesenchymal cells; autologous bone marrow and umbilical cord blood SCs are the most common cell type applied in recent clinical trials. The use of autologous stem cells bypasses the potential for transplant rejection. The harvesting of SCs in the acute stage of brain injury is feasible and safe [61]. Currently reported routes of administration of MSCs may be intravenous and intrathecal. Intrathecal injection may be associated with local tissue injury, whereas intravenous administration may result in cells’ entrapment in areas remote from the injury. The advantage of the intravenous route is its noninvasiveness, but not all infused cells can reach the site of injury. The optimal dose in clinical trials has not yet reached consensus. Similarly, much work is required to determine the proper timing of stem cell treatment. In response to acute TBI, increased proinflammatory cytokines are recruited to the site of damage within 48 h. Stem cells exhibit paracrine secretion of growth factors and cytokines; these can alter cellular immune response to injury to promote neuroregeneration, promoting a favorable functional outcome. However, infusing stem cells into this toxic regional microenvironment too early may not favor engraftment and lead to poor stem cell survival and decreased treatment efficacy. In a clinical study of MSCs, in the acute stage, SCs were administered after 48 h of TBI [62]. To derive maximal benefit, treatment ideally should be offered to those with severe TBI and the worst prognosis. Clinical outcomes measured should include decreased acute treatment intensity and duration, and recovery in motor, sensory, cognition and psychological function.

Some limitations may be found in the clinical application of autologous stem cell therapy. A dedicated multidisciplinary team is required to be accessible. This allows prompt harvesting, preparation and infusion of autologous stem cells. Medical facilities involved also need to be equipped with infrastructure allowing early identification of all TBI patients, and a neurotrauma critical care unit and team are also essential. These are not readily available at every medical facility and thus may limit clinical application of autologous stem cell therapy.

Various animal studies investigated methods to improv transplanted stem cell survival. These included the concomitant introduction of granulocyte-colony stimulating factor. This offers neuroprotection, optimizing stem cell survival at the site of injury [76]. Erythropoietin also improved the survival of transplanted stem cells, and combination therapy showed more effective healing than separate administration [77].

Newer studies on the application of stem cell-derived secretomes and exosomes for TBI were performed. Secretomes are composed of cytokines, chemokines, growth factors, proteins, lipids, nucleic acids, metabolites and extracellular vesicles. Secretomes derived from MSCs show promising protective results in TBI, suggesting potential for future treatment exploration [78]. Exosomes are extracellular vesicles that carry multiple signal moieties, including proteins, lipids, cell surface receptors, enzymes, cytokines, transcription factors and nucleic acids. They can be secreted by a variety of cell types, including MSCs, and aid in a wide variety of cellular functions. They are favored for tissue engineering and regeneration due to their advantageous properties of stability, biocompatibility, low toxicity and proficient exchange of molecule contents [79]. The effectiveness of MSCs in promoting neurological functional recovery in TBI has partly been due to secreted exosomes; thus, the delivery of these exosomes has the potential to limit TBI-related adverse effects and restore neurological function [80]. Specific to animal TBI studies, umbilical cord mesenchymal stem cell-derived exosomes were isolated and injected into ventricles of rats with TBI. In TBI, astrocytes and microglia can be excessively activated, resulting in glial scarring and worsened secondary brain injury. The injection of exosomes exerts an inhibitory effect on astrocyte and microglia hyperactivation, thus promoting better neurological outcome [81]. Similarly, intracerebroventricular microinjection of human adipose MSCs into a TBI rat model promoted functional neural recovery, inhibited neuroinflammation and increased neurogenesis [82].

## 6. Limitation

A detailed examination of the molecular mechanisms of stem cell therapy in traumatic brain injury was not covered in this review. Most studies were based on animal models, the relevant information of which is extensive and beyond the scope of this review. This paper focused mainly on a few available clinical studies.

## 7. Conclusions

The SVZ response to TBI is complex, and most of the previous studies are based on animal models. Human responses differ; older human brain seems to elicit a weaker proliferative response. To understand the initial effect of injury, the subsequent response and the mechanism behind age-related differential response, further therapeutic intervention aimed at regenerating damaged neurons and restoring neural function may be more effectively designed. Successful neurogeneses after TBI require not only neural progenital cell proliferation but also the ability to migrate and survive at the appropriate site of injury with eventual differentiation, maturation and incorporation into an existing neural unit.

In bridging from animal to clinical human TBI studies, models of MSC therapy can be safely and effectively engineered. The developing brain in children is particularly vulnerable, and TBI may result in chronic or delayed adverse neurological outcomes. Mesenchymal stem cell therapy can be utilized in acute and chronic settings, aiding neural regeneration, modulating neuroinflammatory response and reducing long-term disability. This dual role of MSC therapy in pediatric TBI has great potential and warrants further clinical studies. The newer, exciting positive effect of mesenchymal-derived secretomes and exosomes in the treatment of TBI also render further studies.

## Figures and Tables

**Figure 1 ijms-24-14706-f001:**
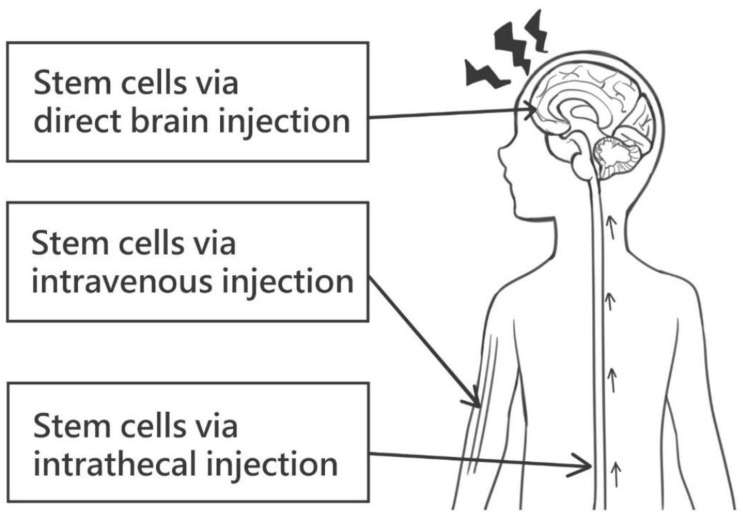
Various routes of stem cell administration. These may include direct brain injection, intravenous injection and intrathecal injection.

**Figure 2 ijms-24-14706-f002:**
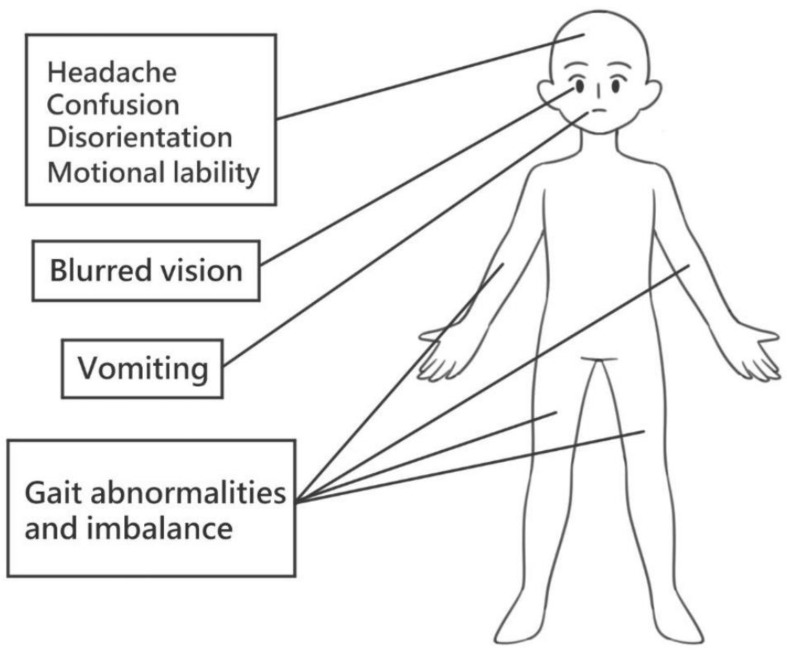
Clinical symptoms following acute TBI. Following acute traumatic brain injury, various clinical presentations such as headache, confusion, disorientation, emotional lability, blurred vision, vomiting and gait abnormalities may be found.

**Figure 3 ijms-24-14706-f003:**
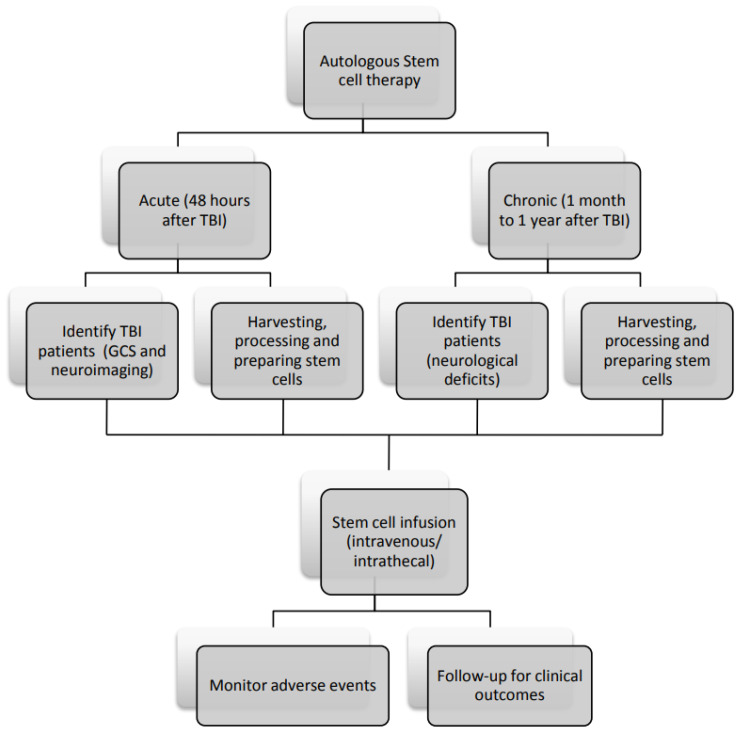
Flowchart of mesenchymal stem cell therapy in TBI patients. This is a basic flowchart demonstrating the clinical application of mesenchymal stem cell therapy in patients identified with traumatic brain injury. Therapy may be initiated in the acute and chronic setting. After harvesting, processing, preparation and infusion of mesenchymal stem cells, patients are then monitored for clinical outcome and adverse side effects.

**Table 1 ijms-24-14706-t001:** Classification of stem cells according to origin and potency.

A. Origin of extraction				
Amniotic cells	Umbilical cord	Bone marrow	Adipose tissue	
embryonic stem cells induced pluripotent stem cells human amnion-derived stem cells		hematopoietic stem cell (lymphoid and myeloid lineage)	adipose-derived stem cells	
B. Potency				
Totipotent/Omnipotent	Pluripotent	Multipotent	Oligopotent	Unipotent
form embryonic (embryo) and extra-embryonic tissue (placenta)	form cells arising from all three germ layers (ectoderm, mesoderm, endoderm)	form cells from single germ layer	form two or more cell lineages within a specific tissue	form only one specific cell lineage type
Most undifferentiated --------------------------------------------------------------------------------------------------------------------------------> Differentiated				

**Table 2 ijms-24-14706-t002:** Main progenitor cell stream types from brain SVZ.

Cell Stream Type	Final Designation	Studied Groups
Rostral migratory stream	Olfactory bulb	Rats
Medial migratory stream	Prefrontal cortex	Humans
Ventral migratory stream	Nucleus accumbens (islands of calleja)	Mice
Ventral migratory stream	Claustrum	Humans
Dorsal migratory stream	Occipital cortex	Mice
Dorsal migratory stream	Upper-layer glutamargic neurons	Mice

**Table 3 ijms-24-14706-t003:** Published human clinical trials of stem cell therapy in patients with traumatic brain injury.

Authors (Reference)	Case Numbers	Age	Timing of Intervention	Stem Cell Source	Route and Dose	Outcomes Measured	Improvement	Followed Time of Improvement	Major Adverse Events
Liao et al. [61]	10 (19 control)	5–14	<48 h of injury	Autologous bone marrow mononuclear cells	Intravenous, 6 × 10^6^ stem cells/kg	Pediatric intensity level of therapy (PILOT score)	Decreased score and treatment intensity for raised intracranial pressure	Day 2 to day 21, significant improvement within week 1	No
						Pediatric logistic organ dysfunction (PELOD score), days of ICP monitoring	Decreased severity of organ injury and days of ICP monitoring	Day 7 to day 21	
Cox et al. [62]	10	5–14	<48 h of injury	Autologous bone marrow mononuclear cells	Intravenous, 6 × 10^6^ stem cells/kg	Functional outcome - Glasgow outcome scale - Glasgow outcome scale—expanded for children - pediatric injury functional outcome scale - adaptive behavior assessment system II	Significant improvement	6 months	No
						Neuropsychological outcome - Wechsler Abbreviated Scale of Intelligence - coding - grooved pegboard - listening recall, verbal learning	Significant improvement		
						Magnetic resonance imaging (MRI) volumetric study	No significant change in grey matter, white matter or intracranial or CSF volume.		
Wang et al. [63]	20 (20 controls)	Adults	>1 year after injury	Umbilical cord mesenchymal stem cells	Intrathecal, 1 × 10^7^ stem cells (4 courses)	Fugl-Meyer Assessment (FMA) - upper and lower extremity motor - sensation - balance	Significant improvement	6 months	No
						Functional Independence Measures (FIM) - self care - mobility - locomotion - communication	Significant improvement		
Tian et al. [64]	97	-	>1 month after injury	Autologous bone marrow stem cells	Intrathecal, 1 × 10^6^ stem cells	Function of brain (39.2%)	Significant improvement	14 days	No
						Consciousness improvement (45.8%)			
						Improved motor functions (37%)			

**Table 4 ijms-24-14706-t004:** Characteristics and factors to be considered in clinical mesenchymal stem cell application.

A: Characteristics	References
Differentiate to cell of neuronal lineage	[66,67]
Migrate to site of injury	[68]
Cross blood–brain barrier	[70,71,72]
Modulate neuroinflammatory response; favor regeneration	[73]
B: Clinical consideration	References
Source Dose Delivery Timing Patient selection Measured outcome	[61,62,63,64]

## Data Availability

Not applicable.

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
