# Peer review of "Stem Cell Therapy in Children with Traumatic Brain Injury"

_ijms, 2023, doi:10.3390/ijms241914706_

Round 1
Reviewer 1 Report
In this review article the authors provide a good introduction into the use of stem cell therapy for pediatric traumatic brain injury. The manuscript is written from a clinical perspective. After a more general introduction into stem cells and traumatic brain injury, the main body of the manuscript is focused on stem cell therapeutic approaches in children. It ends by giving an outlook to ways of introducing this therapeutic model in the clinical practice.
Overall the review article is well written and interesting to read. Nevertheless, there are some points which could be improved by the authors:
1) English language The use of the English language is acceptable in most parts of the manuscript. In some parts, for example in the first paragraph of section 3.3 the number of errors is very high and interferes with an easy reading of the manuscript. Less frequent spelling and grammatical errors, however, are present throughout the manuscript.
2) Paragraph 3.1 This paragraph is a brief introduction into neurogenesis in humans. The question, of whether endogenous neurogenesis or stem cell activation s triggered by TBI is not well addressed (only one paper from 2013 is discussed in this context). This paragraph should discuss much more broadly studies about the induction of endogenous stem cells after TBI.
3) Paragraph 3.2 This paragraph has the title "Peripheral response to injury". This title should be replaced by something like "Immune response and inflammation after TBI". For covering this area, the paragraph is very short and concentrates on a small number of papers. It would benefit from being extended a little bit.
4) Paragraph 3.3 Stem cell therapy In this paragraph most three publications from clinical trials are discussed that used stem cell therapy in children and adults after TBI. These articles are well discussed. However, they were all published between 2011 and 2015, ie. they are all about 10 years old. It is difficult to believe that there are no more recent clinical studies available. The author should at least explain why they refer to these old studies and why no more recent studies are available.
see first point above
Reviewer 2 Report
Pediatric traumatic brain injury causes mortality and severe neurological consequences with long-term morbidity. The authors have based their study on the available literature that stem cell therapy has become a potential new treatment, mainly for adults, and the fact that for children the information is still lacking. Overall, the next research must find a mechanism of action where an effective stem cell therapy is used, the type of stem cell therapy, optimal timing, combination with other treatments, and selection of target group or patients. The authors here have focused and presented stem cell therapy in children with traumatic brain injury.
This review summarizes what authors have identified in the literature. There are a couple of points that need to be clarified:
1. This review is not a systematic review as the authors have not followed the PRISMA and adhere to what is recommended for the systematic reviews. However, the authors must state what type of this review is, narrative, focused, topical, etc. And how the readers can ensure that the available information is collected completely and without ignorance of the main findings in the literature. Please add a search strategy in case a change of bias is there due to the lack of a systematic method for literature search.
2. Please add the limitations of this review
3. Please add the limitations of cell therapy and the potential safety and toxic side effects.
4. Would that be respondents and non-responders to this type of therapy? If so which factors can predict a responder?
5. Perhaps it is a good idea to add a diagram or schematic of how this technique of cell therapy is used for treatment in children. how the authors see the overview of the process.
6. Would the sex of the patient or age influence the results?
Please proof read the text.
Reviewer 3 Report
In this work, the authors reviewed the stem cell therapy possibilities and results in children with traumatic brain injury.
The idea of this study - is interesting; nevertheless, this manuscript needs some improvements and corrections before publishing may be possible.
General points:
Please add a list of abbreviations before References section to your manuscript.
Special points:
For better readability please add at least two Figures to your manuscript.
Please add in the whole text the multiple references after each sentence – too few references in the whole manuscript.
All tables: please correct the layout of all tables in your manuscript, please do it more accurate.
Please add the references for the Table 4.
Please do your References List according to IJMS.
Round 2
Reviewer 3 Report
Thank you for all corrections.
Author Response
Thank you for your comments.